# Proteinuria and Electrophoretic Pattern in Dogs with Comorbidities Associated with Chronic Kidney Disease

**DOI:** 10.3390/ani13081399

**Published:** 2023-04-19

**Authors:** Alicia Pamela Pérez-Sánchez, Sofía Perini-Perera, Javier Del-Angel-Caraza, Israel Alejandro Quijano-Hernández, Sergio Recillas-Morales

**Affiliations:** 1Hospital Veterinario para Pequeñas Especies de la Facultad de Medicina Veterinaria y Zootecnia-Universidad Autonoma del Estado de México, Toluca 50130, Mexico; 2Hospital Veterinario de Pequeños Animales, Facultad de Veterinaria, Universidad de la República, Montevideo 13000, Uruguay

**Keywords:** proteinuria: creatininuria ratio, chronic kidney disease, SDS PAGE, renal biomarker, dogs

## Abstract

**Simple Summary:**

The detection of proteinuria allows for the early diagnosis of chronic kidney disease. This work was to identify and determine the magnitude of proteinuria and its electrophoretic pattern in dogs with chronic diseases related to proteinuria. Data from 264 dogs were obtained; proteinuria was observed in more than 30% as the only finding of kidney disease, evidencing a greater risk factor for proteinuria in the heart disease group. A higher frequency of glomerular electrophoretic patterns related to glomerular hypertension was observed in the heart, neoplasia and endocrine disease.

**Abstract:**

In animals with chronic pathologies, the detection of proteinuria via the proteinuria: creatininuria ratio (UPC) and urinary protein electrophoresis allows for the early diagnosis of chronic kidney disease (CKD). The objective of this work was to identify and determine the magnitude of proteinuria and its electrophoretic pattern characterization in dogs with chronic diseases pathophysiologically related to proteinuria. With the studied patients, five groups were formed. The control group (CG) contained non-proteinuric cases. The cases with proteinuria were classified into four groups according to the concurrent disease: chronic inflammatory diseases (IG), neoplasms (NG), heart diseases (HG), and endocrine diseases (EG). For the statistical analysis, descriptive statistics and non-parametric tests were used. Data from 264 dogs were obtained; in the disease groups, proteinuria was observed in more than 30% as the only finding of kidney disease, evidencing a greater risk factor for proteinuria in the HG group (OR 4.047, CI 1.894–8.644, *p* < 0.0001). In the HG, NG, and EG groups, a higher frequency of glomerular pattern (GEP) related to glomerular hypertension was observed; in the IG, a higher frequency of mixed pattern (MEP) was observed. These findings are secondary to the hyperfiltration process that affects the glomerulus and the renal tubule.

## 1. Introduction

Proteinuria is defined as the abnormal excretion of protein in the urine and is caused by an alteration in the function of the nephron [1]. The glomerular capillary wall, which is formed by the fenestrated endothelium, the glomerular basement membrane, and the podocytes with their slit diaphragm, constitutes a filter allowing the passage of substances depending on their molecular weight (regularly < 70 kDa) and ionic charge, limiting the substances that become filtered; only a small amount of albumin passes through the filter and is largely reabsorbed in the proximal tubule [2]. Various alterations in nephron function can lead to chronic protein loss in the urine, resulting in an overload of the physiological mechanisms responsible for reabsorbing filtered proteins, which generates damage to tubular cells; in its chronic course, it leads to fibrosis, interstitial, tubular degeneration, and nephron atrophy [3].

Proteinuria can occur from the early stages of chronic kidney disease (CKD), representing an early marker of disease, to advanced stages, where it is considered a prognostic indicator [1,4]. The determination of its magnitude and persistence is carried out through the proteinuria: creatininuria (UPC) ratio, which is a fundamental part of the diagnostic and staging process of CKD [5]. Therefore, an early intervention by the clinician when detecting the magnitude and persistence of CKD allows the establishment of therapeutic strategies to improve the patient’s quality of life and increase survival [6,7].

To recognize the location of the damage in the nephron, a renal biopsy and histopathology of the tissue obtained are the methods of choice, allowing the severity of the damage to the glomerulus, tubule, or interstitium to be determined. Unfortunately, in the veterinary clinic, this procedure is impractical [3]. Therefore, the use of simple and minimally invasive techniques is recommended, such as sodium dodecyl sulfate-polyacrylamide gel electrophoresis (SDS-PAGE) [8]. This technique separates proteins based on molecular weight, allowing the location of the damage to be identified in either the glomerulus, the tubule, or both [9].

Glomerular proteinuria occurs when the selective permeability of the glomerular basement membrane is altered and is characterized by the excretion of medium-molecular-weight (40–69 kDa) and high-molecular-weight molecules (≥70 kDa) [2,9,10]. Damage to the glomerulus is the result of the formation and deposition of immune complexes, causing lesions such as membranous, membranoproliferative, or proliferative glomerulonephritis, glomerular sclerosis, or amyloidosis. In addition, alterations at the glomerular level can develop secondary to acquired systemic processes such as neoplasms, infectious diseases, chronic inflammatory diseases, or heart disease or secondary to endocrine diseases such as diabetes or hyperadrenocorticism [11].

On the other hand, tubular proteinuria is characterized by the presence of low-molecular-weight molecules (<40 kDa) in the urine. Normally, these proteins cross the glomerular filtration membrane freely and are reabsorbed by the proximal tubule. However, when there is a tubular injury, protein reabsorption is affected, generating the urinary excretion of proteins of this type [9]. Causes of this are acute tubular necrosis or Fanconi syndrome, among others [1].

Finally, mixed proteinuria occurs when glomerular and tubular lesions develop simultaneously, showing low-, medium-, or high-molecular-weight proteins in the urine [2,9].

When glomerular proteinuria becomes a chronic process, there are changes in the glomerular basement membrane that generate an increase in the size of the pores, which allows the passage of a greater number of proteins of low molecular weight and, gradually, those of high molecular weight. This increase in the load saturates the absorption mechanisms of the tubular cells, causing decreased reabsorption and, consequently, proteinuria [2,3]. Detecting any of the three types is considered essential for the diagnosis, follow-up, and evaluation of the treatment of patients with CKD [8,12]. The objective was to identify and determine the magnitude of proteinuria and its characterization based on the type of electrophoretic pattern in dogs with different comorbidities associated with chronic kidney disease.

## 2. Materials and Methods

This study is a descriptive, prospective case-control study. It was carried out at the Hospital Veterinario de Pequeñas Especies de la Facultad de Medicina Veterinaria y Zootecnia de la Universidad Autónoma del Estado de México in Toluca City-Mexico, during the period from August 2016 to July 2018.

Dogs that were admitted through the internal medicine area and presented pathologies related to proteinuria were evaluated, obtaining groups with chronic inflammatory diseases (IG), neoplasms (NG), heart diseases (HG), and endocrine diseases (EG). Each animal underwent a physical examination, and data on breed, sex, age, body condition score (BCS), and previous diagnoses were considered. Complementary studies were carried out, such as the determination of hormones, dermatological tests, radiographs, ultrasound, and cytology of neoplasms, among others, to arrive at the definitive diagnosis of the associated comorbidity in each case.

To identify the presence of alterations in renal function concomitant to the diagnosed pathologies, each patient underwent the CKD diagnostic protocol suggested by IRIS [5]. After an 8-h solid fast, a blood sample was taken from the jugular vein to perform a complete blood count (ProCyte, IDEXX Distribution Inc. Westbrook, ME, USA), a biochemical profile of 20 analytes (Catalyst One, IDEXX Distribution Inc. Westbrook, ME, USA), and SDMA determination by mass spectroscopic liquid chromatography (IDEXX Laboratories, Inc. Westbrook, ME USA). Urine samples were simultaneously taken by cystocentesis to perform urinalysis and quantitative and qualitative determinations of protein in the urine. The complete urinalysis included the determination of the urine specific gravity (USG) by refractometry, the chemical examination using the reactive strip (VetLab UA, IDEXX Distribution Inc. Westbrook, ME, USA), and the microscopic examination to determine the cellularity of the urinary sediment. When observing inactive urinary sediment, the UPC was determined by spectrophotometry using a semi-automatic analyzer BTS-350 (BioSystem, Barcelona, Spain). For the quantitative determination of urinary proteins, the method of pyrogallol red and molybdate in the acid medium, and for the measurement of urinary creatinine, the modified Jaffé method (picrate in alkaline medium) was used. With the data obtained from the UPC, patients were classified as non-proteinuric (NP), with a value < 0.2, borderline proteinuric (BP), with a value between 0.2 to 0.5, or proteinuric (P), with a value > 0.5 [7]. 

The type of proteinuria was qualitatively identified by SDS-PAGE, using 10% polyacrylamide separating gels and 5% stacking gels [13]. For each patient, the electrophoretic pattern was determined, according to the molecular weight of the protein bands observed in the gels. Protein bands ≥ 40 kDa in the gels were considered glomerular electrophoretic pattern (GEP), and protein bands < 40 kDa were considered tubular electrophoretic pattern (TEP). Bands with low-, medium-, or high-molecular-weight, were considered as mixed electrophoretic pattern (MEP); since no band was observed in the lane, it was considered negative (Figure 1) [8,9].

Blood pressure was measured indirectly using an oscillometer (Vet 20, SunTech Medical Inc. Morrisville, NC, USA), according to the protocol described by Acierno et al. [14]. Systolic blood pressure (SBP) values < 140 mmHg were considered as normotension (NT), 140–159 mmHg as prehypertension (PH), 160–179 mmHg as arterial hypertension (HTN), and ≥180 mmHg as severe hypertension (SH).

The cases considered of interest were those where a UPC > 0.2 was evidenced and/or an alteration of the glomerular filtration rate and/or some grade of arterial hypertension. All clinical cases with a diagnosis of acute kidney injury, CKD with a previously established therapeutic management, and those with active sediment were excluded from this study. 

With the information obtained, five groups were formed. The control group (CG) was formed by animals with a diagnosis of some comorbidity without evidence of alterations in renal function at the time of the diagnostic process. The study groups were made up of animals with some comorbidity, including the chronic inflammatory diseases group (IG), the neoplasm group (NG), the heart disease group (HG), and the endocrine disease group (EG), together with alterations in the evaluation of renal function [15], such as a USG below the critical point and a UPC > 0.2 and/or alteration of the GFR, determined by sCr ≥ 125 μmol/L, and/or SDMA ≥ 15 μg/dL.

Statistical analysis was performed with Graph Pad Prism, version 9.0 (California, USA, 2018). A Kolmogorov-Smirnov normality test was performed, and due to the non-normal distribution of the data, non-parametric tests were used for statistical analysis. Descriptive statistics were used for sex, race, age, BCS, USG, sCr, SDMA, urea, phosphorus, albumin, UPC, electrophoretic pattern, and SBP. A univariate analysis was performed to determine risk factors among groups. The Kruskal-Wallis test was applied to determine differences among the study groups, followed by a post hoc Dunn’s multiple comparisons test. Statistical significance was considered at *p* < 0.05.

## 3. Results

### 3.1. General Distribution of the Population

A total of 388 animals were analyzed, of which 264 met the established study criteria, and 45.8% (n = 121) were females. The median age of the population was 10 years old, with a range from 1 to 18, with 90.5% being older than 6 years. Thirty-four different breeds were found; the most frequent ones were Poodle with 17.0% (n = 45), Miniature Schnauzer with 16.3% (n = 43), Labrador with 6.8% (n = 18), Golden Retriever with 6.4% (n = 17), and Chihuahua with 6.0% (n = 16). In addition, 46 Mongrel dogs were observed.

### 3.2. Distribution of the Groups

The CG was made up of 107 animals (40.5%), whereas the study groups were made up of a total of 157 cases. The IG contained 64 cases (24.3%), the NG 56 (21.2%), the HG 25 (9.5%), and the EG 12 cases (4.5%). A univariate analysis was performed to identify risk factors among groups (Table 1). The animals with cardiac conditions presented a higher probability of alterations in renal function compared to the other groups of pathologies (OR 4.047, CI 1.894–8.644, *p* < 0.0001).

### 3.3. Differences among Groups

Table 2 shows the medians and percentiles for the variables analyzed. The age variable presented differences between the CG and NG and between the CG and HG, with the CG having a lower median age compared to the latter two groups. In addition, the median age of the EG (9.5 years) was lower than that of the NG.

Regarding BCS, the CG and EG animals presented a higher BCS compared to the IG, NG, and HG. In terms of USG, the CG differed from the IG and NG, with the median value being lower in the latter two groups.

For sCr, statistical differences were observed between the CG and NG, with the median value being higher in the NG. In turn, lower medians were observed in the NG and GE and higher ones in the IG and HG. Regarding SDMA, there were differences between the CG and the four treatment groups.

For phosphorus, the CG differed from the NG and EG, with higher median values in the latter two groups. In addition, the EG had a higher median than the HG. For albumin, a lower median was observed in the NG compared to the CG. Finally, SBP did not differ among the groups.

### 3.4. SDMA, UPC and sCr Values in All Groups

In all groups, the distribution of renal biomarkers was characterized. For the IG, 39% of the animals exclusively presented proteinuria, 25% presented only increased SDMA, and 14% presented proteinuria and increased SDMA simultaneously. The remaining 22% presented azotemia. Regarding the NG, 30.4% of the animals presented proteinuria, 21.4% presented only increased serum SDMA, 16% presented proteinuria and increased SDMA simultaneously, and the remaining 32.2% of the cases were classified as azotemic. In the HG group, 20% of the animals exclusively had proteinuria, 64% had increased SDMA alone, and 8% had proteinuria and increased SDMA simultaneously. In terms of the EG, 33.3% of the animals presented proteinuria as the only finding, 25% presented an increase in UPC and SDMA together, and 41.7% of the cases were azotemic.

### 3.5. UPC, Electrophoretic Pattern, and SBP in All Groups

Table 3 shows the frequency distribution of UPC, electrophoretic pattern, and the degree of hypertension for the groups. The severity of UPC was compared among the study groups (Table 3), and animals of the IG, NG, and EG presented a higher frequency of UPC values > 0.5. In contrast, the HG showed a higher frequency of BP. Significant statistical differences were observed between the CG and the other study groups (Figure 2), with the median UPC being higher in the study groups. The Kruskal-Wallis test (*p* < 0.0001) and the post hoc Dunn test were performed.

In the CG, some electrophoretic pattern was observed in 77.6% (83/107). The dogs of the NG, HG, and EG presented a higher frequency of glomerular pattern, whereas the cases of the IG presented a higher frequency of mixed pattern. Hypertension occurred in the animals of the IG and NG, with a frequency of 7% (11/157) (Table 3).

### 3.6. Ratio of UPC to Electrophoretic Pattern

When analyzing the behavior of the UPC with respect to the electrophoretic pattern, it was observed that the TEP, GEP, and MEP patterns presented a higher median with respect to the negative pattern, which was 0.1 (0.02 to 0.45), cataloged in the range of non-protein patients and borderline proteinuria; animals with TEP had a median of 0.5 (0.03–1.8), those with GEP had a median of 0.17 (0.003–2.7), and those with MEP had a median of 0.17 (0.03 to 2.74) (Figure 3). For both GEP and MEP, five animals with a UPC value > 2.0 were observed for each pattern type.

### 3.7. Relationship of UPC to SBP

In Figure 4, the UPC value is shown relative to the blood pressure value. The median values of UPC were as follows: 0.41 for NT animals, 0.59 for PH animals, 1.7 for HTN animals, and 3.9 for HS animals. Regarding the relationship between SBP and the type of electrophoretic pattern of the groups, no significant differences were observed (*p* = 0.66).

## 4. Discussion

Regarding the age of the population, the median of the different study groups was 9 years old or more, an age range where chronic degenerative pathologies are frequent [16]. The most frequently found breeds were Poodle, Schnauzer, Labrador, Golden Retriever, and Chihuahua. This differs from the findings of O’Neill et al. [6], where the most affected breeds were Yorkshire Terrier, Jack Russell Terrier, and West Highland White Terrier; this difference can be attributed to the preferences of the owners for some races in the places where this type of study was carried out.

For the distribution of the groups, the IG was characterized by the presence of chronic inflammatory diseases, such as periodontal disease, dermatopathies, musculoskeletal disorders, and infectious diseases (leptospirosis, ehrlichiosis), coinciding with what has been reported by other authors. For example, O’Neill et al. [6] determined that 30.3% of the studied population of dogs with risk factors for CKD had periodontal disease, and 24.6% had musculoskeletal disorders. In addition, Pelander et al. [16] observed that inflammatory pathologies were related to higher morbidity and mortality in CKD.

For the NG, the most frequently diagnosed tumors were breast adenocarcinoma, mast cell tumor, soft tissue sarcoma, lymphoma and lipomas, among others, similar to the findings of Prudic et al. [17]. Neoplasms can cause structural changes in the glomerulus, resulting in proteinuria [15]. According to Crivellenti et al. [18], who conducted a study to determine glomerular alterations in bitches with mammary gland carcinoma, a third of the bitches were proteinuric and presented renal structural alterations determined by biopsy, in addition to glomerular IgM deposits. This assures us that neoplasms are a risk factor for developing CKD, which can be diagnosed in these patients from early stages; therefore, the evaluation of proteinuria in cancer patients should be performed routinely [17].

Chronic mitral valve disease was mostly diagnosed in the HG. This pathology is the most common cause of heart failure in small-breed dogs, which favors the presentation of cardiorenal syndrome. This syndrome accelerates the progression of renal dysfunction since it involves reduced renal perfusion, congestion of the organ, and neurohormonal changes related to chronic sympathetic stimulation, such as the production of norepinephrine, angiotensin, and endothelin and the release of natriuretic peptide and oxide nitric [19,20]. In this study, cardiac pathologies were a risk factor for CKD, as reported by other authors [6,21], with alterations in serum and urinary biomarkers as a result of cardiorenal syndrome.

Finally, in the EG patients with diabetes mellitus, hyperadrenocorticism and hypothyroidism were observed. It is worth mentioning that in the first two pathologies, there are reports of their close relationship with CKD by causing proteinuria. In animals with diabetes, lesions similar to human diabetic nephropathy have been observed, such as thickening of the glomerular and tubular basement membranes, diffuse glomerulosclerosis, and interstitial expansion by extracellular matrix material, resulting in proteinuria [22]. Regarding hyperadrenocorticism, the presence of glomerulonephritis and glomerulosclerosis has been observed in this pathology, which causes proteinuria and can trigger systemic hypertension [23]. Although there is only one report of a dog with hypertension, which was secondary to CKD, the role of this pathology in the development or its interaction with arterial hypertension is not clear [14], but possible atherosclerosis due to hypercholesterolemia has been suggested [24].

In the different study groups, a significant percentage of exclusively proteinuric animals was observed, followed by animals with an exclusive increase in SDMA and, finally, an increase in both biomarkers simultaneously. Proteinuria occurs from the early stages of CKD, before the increase in biomarkers of glomerular filtration rate, such as SDMA or sCr [1]. The use of early CKD biomarkers, such as UPC, allows its diagnosis in pre-azotemic stages and provides information regarding the possible evolution of the disease as it is an indicator of disease progression [25]. In addition, combining the electrophoresis of urinary proteins allows a more complete perspective of the severity and location of the damage [9]. Especially in patients with comorbidities, which may represent a factor in the onset of kidney disease, early management of these pathologies and proteinuria is considered the mainstay of treatment since it will slow down the progression of CKD [7,26]. Therefore, the need for an evaluation of the complete renal profile according to IRIS [5] is highlighted in animals with chronic comorbidities.

Regarding UPC, 50% of the animals of the IG, NG, and EG were proteinuric, associated with damage to the glomerulus, detected by electrophoresis. For the HG, 60% of the animals presented BP, which could later evolve into proteinuria due to disease progression [1,4].

The most frequent electrophoretic pattern observed in the NG, HG, and EG was GEP, whereas only 13.3% (21/157) of the cases included in the study groups had a UPC value of 2.0 or higher, considered the representative value of glomerular damage [1,7,27]. This could be considered a utility of electrophoresis in the diagnosis of CKD, making it possible to identify the presence of glomerular damage in initial or less severe stages, where, despite a UPC < 2.0, damage is present at the glomerular level.

In the IG, a higher frequency of MEP was observed, attributed to the fact that chronic inflammatory diseases can cause lesions at the level of the basement membrane, endothelium, and podocytes, allowing the passage of larger proteins and causing glomerular proteinuria; when passing in the glomerular filtrate, they accumulate at the level of the tubules, saturating the capacity of the tubule to reabsorb proteins. Therefore, the ability of lysosomes to metabolize proteins is affected, causing damage to the cytoplasm, cell death, and subsequent fibrosis. The passage of medium- and high-molecular-weight proteins through the damaged glomerulus and the impossibility of reabsorbing proteins at the tubular level can give rise to mixed proteinuria [9,28].

The presence of some type of electrophoretic pattern in the CG is due to the fact that some fractions of albumin can be detected in urine or complete albumin in animals without kidney disease, as well as the excretion of proteins such as transferrin, albumin, fractions of albumin, and β 2-microglobulin in healthy animals with UPC values < 0.2 [1].

Hypertension was only present in the IG and NG; in dogs, it is mostly considered secondary to pathological processes, such as CKD, which may occur in these cases as a result of the damage generated by proteinuria and the activation of neurohormonal systems [14,29].

Based on the literature, a UPC value > 2.0 corresponds to damage at the glomerular level (GEP), and a UPC < 2.0 can occur in both glomerular and tubular lesions [1,7]. In our study, cases with a UPC > 2.0 were related to both GEP (3.6%) and MEP (6.8%), which suggests progressive chronic damage of the remaining nephron [4], starting with glomerular damage and progressing with tubular, evidencing MEP [3].

The relationship between UPC and SBP is described as a continuation. There was a directly proportional relationship between the UPC value and the degree of hypertension; the higher the degree of hypertension, the higher the UPC [10]. Due to the loss of functional nephrons, vasodilation of the afferent arteriole occurs, which leaves the glomerular capillary exposed to changes in perfusion pressure. This causes glomerular hypertension, which is aggravated by the activation of the renin-angiotensin-aldosterone system, resulting in the vasoconstriction of the efferent arteriole. Glomerular hypertension is a condition whose only evidence is often the presence of proteinuria and a slight decrease in plasma proteins secondary to excessive loss of albumin due to hyperfiltration. These processes cause alterations in the glomerular basement membrane, flattening of the podocytes, capillary rarefaction, and glomerulosclerosis [10]. Together, hypertension decreases the blood flow in the peritubular capillary, causing tubular hypoxia, which decreases the capacity for tubular reabsorption. This pathophysiological process is described for CKD in dogs, and it may be secondary to comorbidities that favor its presentation [29,30].

## 5. Conclusions

Chronic-degenerative pathologies, such as heart disease, are frequently related to CKD, where proteinuria can occur from the very early stages of the disease, even before the increase in GFR biomarkers, which was observed in more than 30% of the patients studied.

The glomerular electrophoretic pattern appeared more frequently in neoplastic, cardiac, and endocrine comorbidities, associated with inflammatory changes caused in the glomerulus or by glomerular hyperfiltration, unlike in animals with chronic inflammatory diseases, where a mixed electrophoretic pattern was evidenced. Within the limitations of this study, we consider that the follow-up of such cases is necessary to evaluate the persistence of proteinuria. Future studies are needed to evaluate animals with CKD and its relationship with associated comorbidities and the progression of the nephron lesion attributed to proteinuria and its possible correlation with histopathological studies.

## Figures and Tables

**Figure 1 animals-13-01399-f001:**
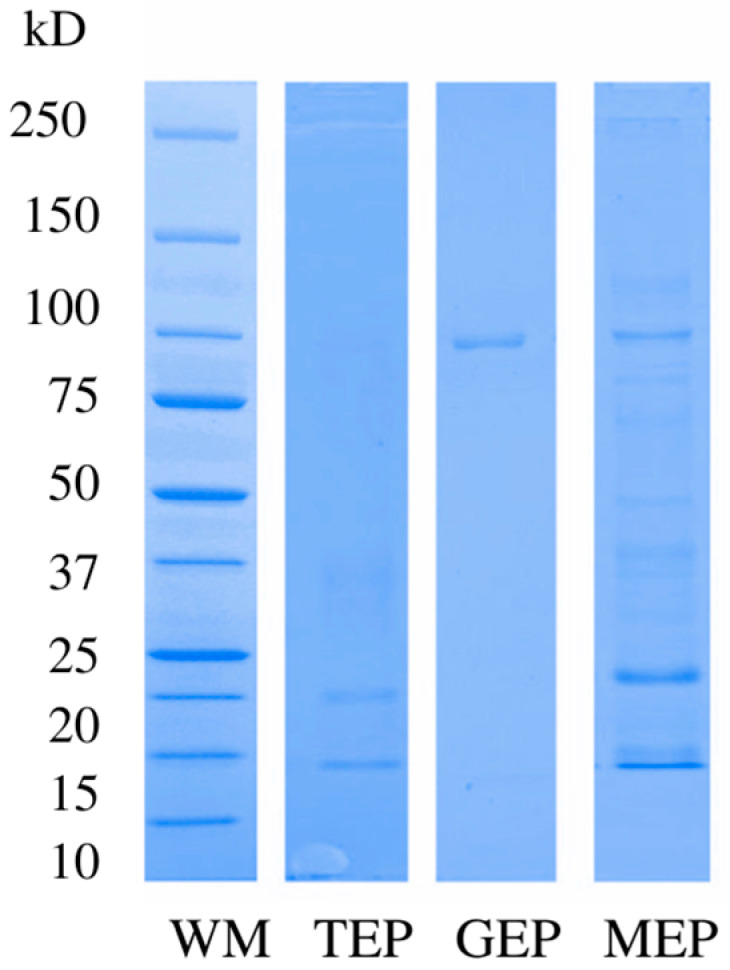
Electrophoretic patterns. Abbreviations: WM, molecular weight marker; TEP, tubular electrophoretic pattern; GEP glomerular electrophoretic pattern; MEP, mixed electrophoretic pattern.

**Figure 2 animals-13-01399-f002:**
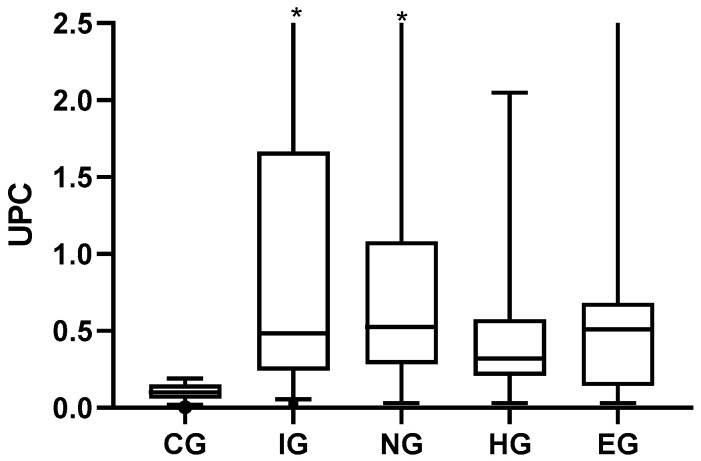
UPC values for the different groups of pathologies. Minimum, maximum, and quartile 1 and 3 in the scatter box plot. Abbreviations: CG, control group; IG, inflammatory diseases group; NG, neoplastic diseases group; HG, heart disease group; EG, endocrine disease group. The asterisks are values outside the scale established for the table multiplied by three.

**Figure 3 animals-13-01399-f003:**
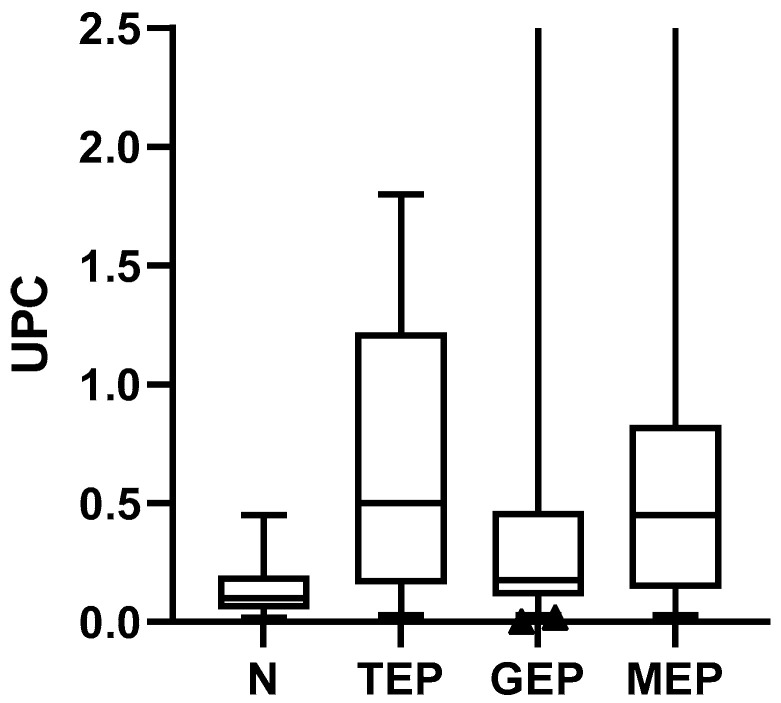
UPC value with respect to the electrophoretic pattern. Kruskal-Wallis (*p* < 0.0001) post hoc Dunn’s test minimum and maximum values are presented, in addition to quartiles 1 and 3. Abbreviations: UPC, proteinuria: creatininuria ratio; N, negative; TEP, tubular electrophoretic pattern; GEP, glomerular electrophoretic pattern; MEP, mixed electrophoretic pattern.

**Figure 4 animals-13-01399-f004:**
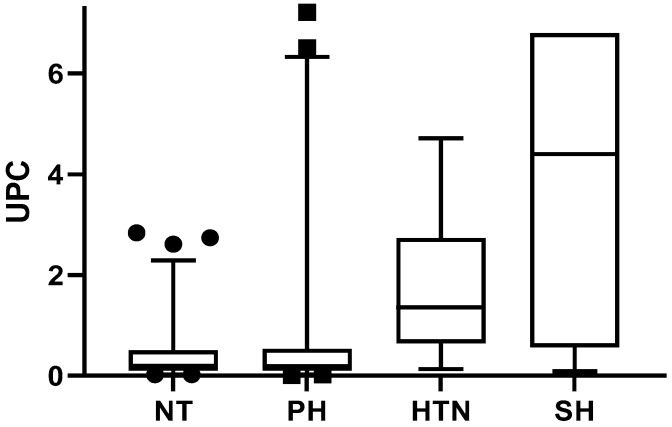
UPC in relation to SBP. Kruskal-Wallis (*p* < 0.0011) post hoc Dunn’s test minimum and maximum values are presented, in addition to quartiles 1 and 3. Abbreviations: NT, normotension; PH, prehypertension; HTN, hypertension; HS, severe hypertension.

**Table 1 animals-13-01399-t001:** ORs calculated for the groups with respect to comorbidities.

Group	CG (n = 107)	Cases per Group (n = 157)	OR	CI 95%	*p*-Value
Inflammatory	54	64	0.675	0.412–1.108	0.11
Neoplastic	37	56	1.049	0.627–1.756	0.85
Cardiac	11	25	4.047	1.894–8.644	**<0.0001**
Endocrine	5	12	1.688	0.577–4.940	0.334

Abbreviations: CG, control group.

**Table 2 animals-13-01399-t002:** Medians and percentiles for age, BCS, USG, sCr, SDMA, urea, phosphorus, albumin, UPC, and SBP for the groups with different pathologies.

Variables N = 264	CG 107	IG 64	NG 56	HG 25	EG 12	*p*-Value
Age (years)	9 (3–14.3) a	9 (3.5–15.4) a, b	11 (5.3–15.6) b	12 (4.6–16.4) b, c	9.5 (5.2–14.7) a, b, c	**<0.0001**
BCS (1/9–9/9)	5 (2–7) a	5 (3–5) b	3 (1–5) b	3 (2–5) b	4 (4–6) a	**0.004**
USG	1.035 (1.005–1.057) a	1.026 (1.008–1.043) b	1.019 (1.007–1.035) c	1.025 (1.010–1.043) a, b, c	1.017 (1.011–1.042) a, b, c	**<0.0001**
sCr (µmol/L)	80 (49.2–122) a	76.5 (39.3–182.0) a, c	88 (4–444.3) b	79 (49.4–173.6) a, c	106 (49.3–477) a, b	**0.031**
SDMA (µg/dL)	10 (5.6–14.0) a	13 (6–49.4) b	15 (7–65) b	13 (6.6–27.8) b	13.5 (11–67) b	**<0.0001**
Urea (mmol/L)	4.8 (1.9–10.0) a	5.7 (1.67–33.4) a, b	5.7 (1.9–46.2) a, b	6.4 (2.8–61.6) a, b	8.1 (4.3–25.7) b	**0.002**
Phosphorus (mmol/L)	1.3 (0.6–1.7) a	1.2 (0.7–3.6) a, b, c	1.4 (0.7–4.6) b, c	1.29 (0.4–1.8) a, c	1.5 (1.2–2.8) b	**0.001**
Albumin (g/L)	29 (21–36.3) a	28 (19.5–34.4) a, b	27 (18–32) b	28 (22–34) a, b	28.5 (23.5–33.4) a, b	**0.002**
UPC	0.1 (0.02–0.19) a	0.48 (0.07–6.63) b	0.52 (0.04–2.78) b	0.32 (0.08–1.54) b	0.51 (0.04–2.20) b	**<0.0001**
SBP (mmHg)	138 (111–158) a	140 (112–178) a	139 (109–172) a	131 (108–157) a	133 (109–153) a	0.214

Abbreviations: CG, control group; IG, inflammatory diseases group; NG, neoplastic diseases group; HG, heart disease group; EG, endocrine diseases group; BCS, body condition; USG, urinary specific gravity; sCr, serum creatinine; SDMA, symmetric dimethylarginine; UPC, proteinuria: creatininuria ratio; SBP, systolic blood pressure. Median and percentiles are in parentheses (0.025–0.975). Different letters indicate significant statistical differences at *p* < 0.05; analysis was performed with Kruskal-Wallis and post hoc Dunn’s test.

**Table 3 animals-13-01399-t003:** Distribution of frequencies and percentages of UPC, electrophoretic pattern, and SBP in the groups studied.

	CG n = 107	IG n = 64	NG n = 56	HG n = 25	EG n = 12	Total N = 264
**UPC**						
NP	1**07** **(100%)**	6 (9.4%)	11 (19.6%)	2 (8.0%)	3 (25.0%)	129 (48.8%)
BP	0 (0%)	27 (42.2%)	**15** **(26.8%)**	15 (60.0%)	2 (16.6%)	59 (22.4%)
P	0 (0%)	**31** **(48.4%)**	**30** **(53.6%)**	8 (32.0%)	**7** **(58.4%)**	76 (28.8%)
**EP**						
N	24 (22.4%)	7 (11.0%)	8 (14.3%)	4 (16%)	0 (0%)	43 (16.3%)
TEP	1 (1.0%)	1 (1.6%)	4 (7.1%)	0 (0%)	1 (8.3%)	7 (2.7%)
GEP	**61** **(57.0%)**	26 (40.6%)	**31** **(55.4%)**	**11** **(44%)**	**7** **(58.4%)**	**136** **(51.5%)**
MEP	21 (19.6%)	30 (46.8%)	13 (23.2%)	10 (40%)	4 (33.3%)	78 (29.5%)
**SBP**						
NT	**58** **(54.3%)**	**32** **(50.0%)**	**29** **(51.8%)**	**16** **(64%)**	**7** **(58.4%)**	142 (53.8%)
PH	49 (45.7%)	29 (45.3%)	19 (33.9%)	9 (36%)	5 (41.6%)	110 (41.7%)
HTN	0 (0%)	1 (1.6%)	7 (12.5%)	0 (0%)	0 (0%)	8 (3.0%)
SH	0 (0%)	two (3.1%)	1 (1.8%)	0 (0%)	0 (0%)	4 (1.5%)

Abbreviations: CG, control group; IG, inflammatory diseases group; NG, neoplastic diseases group; HG, heart disease group; EG, endocrine diseases group. UPC, proteinuria: creatininuria ratio; NP, non-proteinuric; BP, borderline proteinuria; P, proteinuric; EP, electrophoretic pattern; N, Negative; TEP, tubular electrophoretic pattern; GEP, glomerular electrophoretic pattern; MEP, mixed electrophoretic pattern; SBP, systolic blood pressure; NT, normotension; PH, prehypertension; HTN hypertension; SH, severe hypertension.

## Data Availability

The data presented in this study are available on request from the corresponding author.

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
