# Peer review of "Proteinuria and Electrophoretic Pattern in Dogs with Comorbidities Associated with Chronic Kidney Disease"

_animals, 2023, doi:10.3390/ani13081399_

Round 1

Reviewer 1 Report

 There is some point that should be clarified in the materials and methods section.  Specifically at the inclusion / exclusion criteria.

When a patient is assigned to a group, is already receiving therapy or is the moment of initial diagnosis?

I believe it is quite relevant, because glucocorticoids, non steroidal antiinflammatory drugs, agiotensin converting enzime inhibitors, and hypotensive drugs will dramatically affect proteinuria (and renal parameters).  Again, it is important to clarify because it changes the interpretation of research results.

Author Response

Dear Reviewer, we appreciate your review of the manuscript and your objective comments.

We change the wording of the paragraph based on your suggestions (Lines 149-153).

“The cases considered as of interest were those where a UPC > 0.2 was evidenced and/or an alteration of the glomerular filtration rate and/or some grade of arterial hypertension. All clinical cases with a diagnosis of acute kidney injury, or with a diagnosis of CKD with a previously established therapeutic management, and those with active sediment were excluded from this study”.

We appreciate his contributions to improving this manuscript.

Reviewer 2 Report

The study is interesting. The chosen groups are adequate, although the number of those affected by endocrine diseases is low. Especially when diabetes mellitus is a disease closely related to kidney problems.

The paper is not easy to read because there are too many acronyms and many of them are very similar. This also leads to some errors.

* P is phosphorus (table 2) and, in addition, proteinuric (table 3).

* HTN is hypertension (word) (line 255) and third degree of substaging of CKD of IRIS (line 274).

* In figure 4 appears NH (line 300) and in the descriptive text NT (line 303).

* In figure 4 appears SH (line 300) and in the descriptive text HS (line 304).

EP is not a good acronym to refer to an electrophoretic pattern of urinary proteins.

The molecular weight of albumin is 66 kDa (no 70 kDa, as it says online 132), which affects the level from which you can talk about glomerular pattern.

The statement of: “Considering that the weight of albumin is 70 kDa, the finding of the corresponding bands with this molecular weight or higher was considered a glomerular electrophoretic pattern (GEP). Protein bands below 70 kDa were considered as tubular electrophoretic pattern (TEP)” must be justified with more precision, as it is not indifferent if talking about 66 kDa or more or 70 kDa or more, especially if there are proteins very close to 66 and the marker of molecular weight is the one presented in the figure (line 142).

There are some errors in the bibliographic citations

Line 453: 377-385

Line 455: Yalçin, Çetin

Line 460: Quimby

Line 462: Grauer, C.

Lines 462 and 491: Bartges J., Polzin D. In: Bartges, J., Polzin

Line 500: A feline

Line 507: Gewerbestrasse 11, 6330 Cham, Switzerland

Author Response

Dear Reviewer, we appreciate your review of the manuscript and your objective comments.

It is important to comment that Diabetes is not frequent in our population, but all clinical cases admitted during the study period were considered.

In the section on materials and methods, and in the section on electrophoretic patterns, the wording of the text was adapted for better understanding (Lines 129-136)

Regarding your comment on abbreviations, we made the following modifications:

-The abbreviation for phosphorus was removed and the “P” for proteinuria was retained.

-HTN was kept exclusively to substage the third grade according to the IRIS.

-In Figure 4 the error was corrected, and the abbreviation NT was maintained in all cases.

- Removed the abbreviation EP for the electrophoretic pattern.

- For the material and methods part, the wording was modified to stipulate the different electrophoretic patterns, it was poorly written, for which we apologize.

-The errors in the references were modified.

We appreciate all your contributions to improving this manuscript.
